# Shyness and Socio-Emotional Adjustment among Young Chinese Children: The Moderating Role of Screen Time

**DOI:** 10.3390/bs13090763

**Published:** 2023-09-14

**Authors:** Jingjing Zhu, Alicia McVarnock, Laura Polakova, Shuhui Xiang, Yan Li, Robert J. Coplan

**Affiliations:** 1Shanghai Institute of Early Childhood Education, Shanghai Normal University, No. 100 Guilin Rd., Shanghai 200234, China; zhujingjing@shnu.edu.cn (J.Z.); 1000510514@smail.shnu.edu.cn (S.X.); 2Department of Psychology, Carleton University, 1125 Colonel by Drive, Ottawa, ON K1S 5B6, Canada; aliciamcvarnock@cmail.carleton.ca (A.M.); laurapolakova@cmail.carleton.ca (L.P.)

**Keywords:** shyness, socio-emotional adjustment, screen time, young children, China

## Abstract

The primary aim of the present study was to examine the potential moderating role of screen time in the links between shyness and indices of socio-emotional adjustment in young Chinese children. Participants were *N* = 211 children (112 boys, 99 girls) ages 43–66 months (*M* = 58.84 months, *SD* = 5.32) recruited from two public kindergartens in Shanghai, People’s Republic of China. Mothers completed assessments of children’s shyness and screen time, and both mothers and teachers completed measures of indices of children’s socio-emotional functioning (prosocial, internalizing problems, learning problems). Among the results, shyness was positively associated with internalizing problems and negatively associated with prosocial behavior, whereas screen time was positively associated with internalizing problems. However, several significant shyness × screen time interaction effects were observed. The pattern of these results consistently revealed that at higher levels of screen time, links between shyness and indices of socio-emotional difficulties were exacerbated. Results are discussed in terms of the implications of shyness and screen time in early childhood.

## 1. Introduction

Shyness is a temperamental trait characterized by wariness and self-consciousness in social contexts [1] and a well-established risk factor for childhood socio-emotional problems in both Western societies [2,3] and China [4]. However, outcomes related to shyness are known to be heterogeneous [5]. As such, researchers have sought to identify key factors that may determine which shy children are most (or least) at risk of poor socio-emotional functioning. For example, in early childhood, pragmatic language skills were found to attenuate associations between shyness and internalizing problems [6], whereas inhibitory control was found to exacerbate relations between shyness and peer difficulties [7].

Child development is also influenced by the environment [8]. One framework aiming to understand the ways in which temperament and environmental factors interact to predict socio-emotional development is the diathesis–stress model. This model posits that due to underlying diatheses, some children are particularly vulnerable to the negative impacts of environmental stressors [9]. For instance, children high in shyness are especially likely to experience negative outcomes when they are exposed to higher levels of parent neuroticism and overprotective parenting behaviors (e.g., micromanaging of activities) [10,11]. The diathesis–stress framework can be used to expand knowledge regarding why some young shy children adapt better than others to developmental stressors (e.g., transition to early education settings).

With the rise of technology, many children today are exposed to screens from a young age. For the present study, screen time was conceptualized as any engagement with screens, including smartphones, tablets, television, video games, computers, or wearable technology [12]. To optimize children’s developmental trajectories, widely adopted guidelines recommend a maximum of one hour of screen time daily from ages 2 to 5 years [12,13,14,15]. However, cross-cultural evidence suggests that a large majority of preschool-aged children exceed screen time guidelines [16,17], which can have negative socio-emotional and cognitive implications [18,19]. Importantly, young children learn best through face-to-face interactions with family, and time spent engaged with screens takes away from more active opportunities to develop socio-emotional and cognitive skills [12].

To date, researchers have yet to directly examine how adjustment outcomes related to shyness may vary according to screen time usage. However, the implications of excessive screen time may be especially dire for shy children, who, due to socio-evaluative fears and high social reactivity [1], already miss out on important opportunities to build socio-emotional skills through social interaction [20]. Accordingly, drawing upon a diathesis–stress framework, the primary aim of the present study was to examine the potential moderating role of screen time in the links between shyness and indices of socio-emotional adjustment in young Chinese children. Early childhood represents an important and unique developmental period to address this issue, as young children are engaged in the process of developing socio-emotional skills through interactions with peers [21], and parents typically have more control over their children’s technology use. Moreover, early screen time practices may form lasting habits [22], which may help (or hinder) long-term development.

### 1.1. Overview of Shyness in Early Childhood

According to Asendorf’s [23] approach–avoidance model of social withdrawal, shy individuals desire social interactions but are held back by social fears and socio-evaluative concerns. The construct of shyness shares considerable conceptual overlap with related terms, such as behavioral inhibition [24] and social reticence [25]. These terms all share a common theme related to fear and anxiety in social contexts. Shy young children are more likely to forgo opportunities to engage, choosing instead to watch their peers from a distance [25]. Affective, cognitive, and behavioral features of shyness also often mirror those of social anxiety. Although some scholars use the terms interchangeably, research suggests that shyness and social anxiety are distinct constructs [26]. Notwithstanding, childhood shyness remains a significant predictor of the later development of anxiety disorders, particularly social anxiety [27].

Shyness has been linked to a host of socio-emotional difficulties in early childhood. For example, young shy children exhibit deficits in social and communicative skills and are less prosocial than their more sociable agemates [20,28]. It has been suggested that these difficulties reflect, at least in part, deficits in performance rather than deficits in competence [29]. In this regard, young shy children’s empathetic and prosocial tendencies might be overwhelmed by feelings of social fear and socio-evaluative concerns.

Most notably, shy children experience increased internalizing issues, such as loneliness, anxiety, and depression [3,20,30]. Shyness may become particularly problematic during the transition to kindergarten [31]. In kindergarten, children face new social and academic demands, as communication with unfamiliar others (e.g., teachers and peers) becomes more routine and expected. Withdrawing from peers at school may also interfere with the development of strong social skills, which are important for long-term well-being and academic success [32].

Young shy children are less likely to interact with peers or speak up in class [33], and such failure to participate in classroom activities can translate to poor academic adjustment [31]. Indeed, shy children are perceived by teachers as less academically competent [31,32]. Difficulty initiating social interactions can also prevent shy children from developing positive peer relationships. For example, shy young children tend to be viewed more negatively by their peers and experience higher rates of exclusion and victimization [3,34].

### 1.2. Shyness in Mainland China

It is now widely accepted that shyness can have different meanings and implications across different cultures, as cultural values and beliefs are inextricable from human development [35]. For example, in traditional Chinese culture, shyness was viewed as an indication of maturity and social competence that helped to maintain social cohesion in a collectivistic society [36]. In support of this notion, historical studies indicated that shyness in China was associated with positive outcomes, including peer-liking, leadership, and academic success [37]. However, over the last 25 years, there have been large-scale economic and social macro-level changes in China, with individualist factors becoming increasingly important in predicting well-being and success [38,39]. As a result, contemporary research indicates that shyness among Chinese children is now associated with a range of adjustment difficulties, including academic under-achievement, internalizing problems (e.g., loneliness, social anxiety, depression), and negative peer experiences (e.g., exclusion, victimization) [4]. Although most of these studies have focused on older children and adolescents, there have been a handful of recent studies demonstrating similar negative outcomes for shy young children in China [40,41,42,43].

As aforementioned, not all shy children experience adjustment difficulties, which has led to the exploration of risk and protective factors (see Coplan et al., 2020, for a recent review) [5]. A myriad of such factors has been previously examined among Chinese children, including receptive language [43], academic achievement [44], attachment style [45], and teacher–child relationships [42]. However, although there is growing interest in the impact of screen time on child development [19], this construct has not been considered with respect to its potential impact on shy children to date.

### 1.3. Screen Time and Shyness: A Diathesis–Stress Model

Children’s use of screens has increased considerably with advancements in digital technology [46]. For example, by age 5 years, Chinese children spend over 3 h a day exposed to screens, with passive screen time (e.g., television, videos) making up two-thirds of overall exposure [16]. As a result, children are likely to spend less time in social scenarios (including child–parent and peer interactions), which can serve to reduce verbal opportunities and creative play [47].

Excessive screen exposure can impede healthy development in early childhood [46]. For example, studies in both North America and China suggest that young children engaging in high levels of passive screen time are more likely to experience problems with executive functioning and struggle to develop academic, language, and social skills [16,48]. Excessive screen time in childhood is also associated with increased internalizing and externalizing problems, as well as attention impairments [49,50,51]. This may be in part because increased screen exposure often translates to decreased interactions with caregivers and peers [52], which is how young children learn best [12]. As such, children engaging in excessive screen time may have fewer opportunities to develop socio-emotional skills. Moreover, young children engaging in high amounts of screen time may be more likely to overuse technology when they get older [22]. These implications may be particularly relevant for shy children in China.

The diathesis–stress model has been applied to the development of childhood shyness [53] and can be used to conceptualize the interplay between shyness and screen time in early childhood. In this context, shyness is denoted as an individual vulnerability (i.e., diathesis) and screen time as an environmental stressor. From this perspective, social fear and socio-evaluative concerns prevent shy children from engaging with teachers and peers during the transition to kindergarten and hinder the development and implementation of important socio-emotional and language skills [31]. Indeed, even outside of school, young shy children are less likely to engage in playdates and extracurricular peer activities [6,11]. Such restricted opportunities for social interactions may be especially common among shy Chinese children due to the substantive emphasis Chinese families place on academic achievement [44].

Young shy children in China who spend more time on screens (whether passive or active) may further isolate themselves from social engagement. Missed opportunities to interact with others (e.g., caregivers, siblings) and develop friendships may be especially detrimental for young shy children, who may already lack social and language skills [40,41,42,43]. As a result of these diathesis–stress processes, we speculated that links between shyness and socio-emotional maladjustment would be especially strong at higher levels of screen time. There is some support for this notion in older samples, as socially withdrawn adolescents who depend more on their smartphones have been shown to be particularly at risk for interpersonal difficulties [54]. Yet, to date, the impact of screen time on the links between early childhood shyness and indices of adjustment has not been investigated.

### 1.4. The Present Study

To address gaps in the literature, the goal of the present study was to examine links between shyness, screen time, and indices of socio-emotional functioning among young Chinese children. We were particularly interested in the potential moderating role of screen time in the relationship between shyness and adjustment. As aforementioned, previous research has identified links between shyness and maladjustment in three primary domains: (1) internalizing problems, (2) negative peer experiences, and (3) academic difficulties. Accordingly, in the present study, we included a range of outcome variables assessed by mothers and teachers, including indices of internalizing problems (i.e., parent-rated emotional symptoms, teacher-rated internalizing problems), aspects of peer experiences (i.e., maternal-rated prosocial behaviors and peer problems), and academic difficulties (i.e., teacher-rated learning problems). It was hypothesized that, overall, shyness would be positively associated with internalizing problems, negative peer experiences, and learning difficulties, as well as negatively associated with prosocial behavior. Further, in line with diathesis–stress processes, we speculated that links between shyness and socio-emotional functioning would be moderated by screen time, such that associations between shyness and negative outcomes (including reduced prosocial behavior) would be strongest at higher levels of screen time.

## 2. Materials and Methods

### 2.1. Participants and Procedure

Participants were *N* = 211 children (112 boys, 99 girls) ages 43–66 months (*M* = 58.84 months, *SD* = 5.32) recruited from eight classes in two public kindergartens in Shanghai, P.R. China. There were 25–30 children in each class. In China, children attend kindergarten for three years (e.g., junior class: age 3–4 years; middle class: age 4–5 years; senior class: age 5–6 years). All children were of Han ethnicity, which is the predominant ethnic group in China (nearly 97% of the population). A total of 14.8% of mothers and 13.8% of fathers completed high school, 21.9% of mothers and 21.4% of fathers had completed junior college, 47.5% of mothers and 37.2% of fathers had a bachelor’s degree, and 15.8% of mothers and 27.6% of fathers had a postgraduate degree. Maternal and paternal scores were averaged to create an overall measure of parental education, with higher scores representing higher education.

The data were collected in February of 2019. The present study was reviewed and approved by the Ethics Review Board of Shanghai Normal University. In China, fathers still remain predominantly in breadwinning roles [55], and mothers assume the primary caregiving duties [56]. Accordingly, and following protocols from previous studies of child social withdrawal and screen time [57,58,59], we relied upon maternal (as opposed to paternal) reports and teacher ratings as sources of assessment in the present study. With the consent of the kindergarten director, recruitment information was communicated to the children’s mothers by each classroom teacher. All children in the kindergarten, as well as their mothers and teachers, were invited to participate in the study. Mothers and teachers provided written informed consent through the corresponding kindergartens before completing the online questionnaires. The online questionnaire was forwarded to children’s mothers via the teachers. The participation rate was 98%. Teachers received a small honorarium (equivalent to approximately $50) after finishing the questionnaire. Mothers were not compensated. The results of the missing data analysis indicated a range of 0.5 to 7.1% missing for all study variables. Little’s [60] test of missingness indicated that data did not significantly deviate from a missing completely at random pattern (*χ*^2^ = 49.12, *df* = 39, *p* = 0.12).

### 2.2. Measures

#### 2.2.1. Shyness

Mothers completed the Chinese version of the Child Social Preference Scale (CSPS) [41,61]. Of particular interest was the sub-scale assessing *shyness* (7 items, α = 0.87, e.g., “My child seems to want to play with other children, but is sometimes nervous”). Items were rated on a 5-point scale (1–5) from “not true at all” to “very true”. Previous research with young children in China indicates good internal and test–retest reliability, along with criterion, concurrent, and predictive validity of the sub-scale [62].

#### 2.2.2. Screen Time

Mothers reported their children’s screen exposure (e.g., defined for them as “watching television, using a smartphone, computer, or other digital media device”) in hours for each day of the previous week (Monday through Sunday). Given our conceptualization of screen time as a broad construct, scores were aggregated to create a single score of screen time [22,63].

#### 2.2.3. Social–Emotional Adjustment

Mothers completed the Chinese version of the *Strengths and Difficulties Questionnaire* (SDQ) [64,65]. Of particular interest were subscales assessing *prosocial behavior* (5 items, α = 0.71, e.g., “Considerate of other people’s feelings”), *emotional symptoms* (5 items, α = 0.69, e.g., “Many worries, often seem worried”), and *peer problems* (5 items, α = 0.39, e.g., “Picked on or bullied by other children”) [66]. Items were rated on a 3-point scale from 1 (doesn’t apply) to 5 (certainly applies). The Chinese version of the mother-rated SDQ has demonstrated satisfactory internal and test–retest reliability, as well as convergent validity among Chinese children [67].

Teachers completed the Chinese version of the Social Skills Teacher Rating System (SSTRS) [41,68]. Of particular interest was the subscale assessing *internalizing problems* (4 items, α = 0.84, e.g., “Feels anxious in a group”). Items were rated on a 3-point scale from 0 (never) to 2 (always). The SSTRS has been shown to be reliable and valid in young Chinese children [41]. Teachers also completed the *learning problems* subscale of the Chinese version of the Teacher Behavior Rating Scale (TBRS) [69]. This 4-item subscale assesses children’s academic difficulties (α = 0.86, e.g., “Has difficulty in learning”). Items were rated on a 3-point scale from 0 (doesn’t apply) to 2 (certainly applies).

### 2.3. Analytic Strategy

Data analyses were conducted using IBM SPSS (version 22.0). First, gender differences were examined using *t*-tests (boy = 0, girl = 1). Second, Hayes’s PROCESS macro [70], with non-parametric bootstrapping with 5000 resamples, was used to examine the potential moderating effect of screen time on the association between shyness and social–emotional adjustment. Moderation was regarded when the 95% bias-corrected confidence interval (CI) of the interaction term (shyness × screen time) did not include zero [71]. To probe significant interactions, simple slope tests using methods suggested by Aiken and West (1991) [72] were conducted. The Johnson–Neyman (J–N) technique was then applied to estimate regions of significance for the adjusted effects of shyness on social–emotional adjustment variables as a function of screen time.

## 3. Results

### 3.1. Preliminary Analyses

Descriptive statistics and intercorrelations among study variables are displayed in Table 1. Boys experienced higher learning difficulties than girls (*M*_boys_ = 0.55, *SD* = 0.54; *M*_girls_ = 0.31, *SD* = 0.48; *t* = 3.28, *p* < 0.001). There were no gender differences in shyness (*M_boys_* = 1.91, *SD* = 0.65; *M_girls_* = 1.80, *SD* = 0.64; *t* = 1.13, *p* = 0.26), screen time (*M*_boys_ = 0.39, *SD* = 0.40; *M*_girls_ = 0.45, *SD* = 0.50; *t* = −0.90, *p* = 0.37), prosocial behavior (*M*_boys_ = 2.74, *SD* = 0.29; *M*_girls_ = 2.81, *SD* = 0.35; *t* = −1.50, *p* = 0.14), emotional symptoms (*M*_boys_ = 1.41, *SD* = 0.40; *M*_girls_ = 1.45, *SD* = 0.48; *t* = −0.66, *p* = 0.51), peer problems (*M*_boys_ = 1.37, *SD* = 0.27; *M*_girls_ = 1.34, *SD* = 0.32; *t* = 0.73, *p* = 0.47), or internalizing problems (*M*_boys_ = 0.14, *SD* = 0.35; *M*_girls_ = 0.10, *SD* = 0.30; *t* = 0.88, *p* = 0.38).

Parent education was negatively associated with screen time, peer problems, and learning problems. Shyness was positively associated with emotional symptoms and peer problems and negatively associated with prosocial behavior. Screen time was positively associated with emotional symptoms. Accordingly, we controlled for parent education and child gender in subsequent analyses.

### 3.2. Shyness, Screen Time, and Social–Emotional Adjustment

Classroom intra-class correlations (ICC) were less than 0.04 and non-significant for all variables, indicating no classroom-based cluster effects. Models examining shyness × screen time interactions on socio-emotional adjustment were conducted separately for mother-rated prosocial behavior, mother-rated internalizing problems, teacher-rated internalizing problems, and learning problems while controlling for parent education and child gender. Results are displayed in Table 2. There were significant shyness × screen time interaction effects for peer, internalizing, and learning problems.

Following suggestions by Hayes and Matthes (2009) [73], we used the Johnson–Neyman (J–N) technique [74] to further probe to determine the cut-off values. All predictors were standardized for the analyses, then a “region of significance” was estimated for the simple slope of a predictor conditioned on the value of the continuous moderator. The cut-off point is where the “region of non-significance” changes to the “region of significance”. This technique allowed us to estimate a region of significance for the simple slope of a predictor conditioned on the value of the continuous moderator. The results are presented visually in Figure 1, Figure 2 and Figure 3.

When screen time was higher than −0.42 SD, shyness was positively associated with peer problems (Figure 1). However, when screen time was lower than −0.42 SD, shyness was no longer associated with peer problems. When screen time was higher than 1.09 SD, shyness was positively associated with internalizing problems (Figure 2). However, when screen time was lower than 1.09 SD, shyness was no longer associated with internalizing problems. When screen time was higher than 1.28 SD, shyness was positively associated with learning problems (Figure 3). However, when screen time was lower than 1.28 SD, shyness was no longer associated with learning problems.

## 4. Discussion

The goal of this study was to assess links between shyness, screen time, and indices of socio-emotional functioning among young children in mainland China. Overall, results indicated that both shyness and screen time were linearly related to indices of socio-emotional difficulties. However, several significant shyness × screen time interaction effects were observed. Consistent with postulations of the stress–diathesis model pertaining to shyness [53], the pattern of these results consistently revealed that at higher levels of screen time, links between shyness and indices of socio-emotional difficulties were exacerbated.

### 4.1. Shyness and Screen Time in China

Overall, results from bivariate correlations indicated that shyness was positively associated with emotional symptoms and peer problems and negatively associated with prosocial behavior. These findings add to the growing number of recent studies linking early childhood shyness in China with internalizing issues (parent-rated) and poor peer experiences [40,41,42,43,62]. Such findings also reinforce the notion that because of changing social norms in China, as is the case in the West, childhood shyness is now a maladaptive trait and carries a risk for negative adjustment outcomes [35,38].

In line with previous evidence suggesting that screen time is on the rise among children in China [16], children in the present study engaged in almost three times the recommended guidelines for screen time. Excessive screen exposure has been linked to a range of negative socio-emotional outcomes, as well as cognitive and academic difficulties, among young Chinese children [49,75]. However, with the exception of a significant (albeit modest) positive association with maternal-rated emotional symptoms, screen time was not linearly related to socio-emotional difficulties. One explanation for the discrepancy between past and present findings relates to differences in how children engage with screens. For example, Hu et al. [16] found that although passive screen time (e.g., television) undermined social, cognitive, and academic capabilities in a sample of Chinese 5-year-olds, active screen time (e.g., educational apps) enhanced science and language performance and was unrelated to social capabilities. Other research suggests that when content is educational and age-appropriate, even watching television can incur socio-emotional and academic benefits for young children [76]. Thus, the link between screen exposure and early childhood adjustment may not be straightforward. We did not differentiate between active and passive forms of screen time (or consider differences in screen content), making this an important area for future research.

Notwithstanding, screen time was found to significantly moderate associations between shyness and adjustment difficulties across several domains (i.e., learning problems, peer problems, internalizing problems). The pattern of interaction effects was similar across all outcomes: as expected, higher levels of screen time heightened associations between shyness and adjustment problems. These findings add to the empirical support for diathesis–stress effects related to shyness [53], suggesting that children higher in shyness are more vulnerable to the effects of screen exposure. Some research suggests that excessive screen exposure during the early years can limit opportunities for peer interaction, which may, in turn, lead to socio-emotional difficulties. For example, Putnick et al. [77] recently found that increased screen time predicted deficits in communication and social skills over time, not directly but indirectly, through decreased peer play. As such, engaging in too much screen time may be particularly problematic for shy children, as it carries the risk of further exacerbating social withdrawal and interfering with the development and implementation of social skills. Perhaps not surprisingly, then, low screen time attenuated the positive association between shyness and peer problems in this study. Findings suggest that limiting screen time may protect against (at least some of) the negative implications of shyness in early childhood.

Although shyness was not directly related to learning difficulties, the combined effect of high shyness and high screen time on learning is noteworthy. Shyness has been previously linked to academic difficulties and reduced vocabulary in early childhood [7,32]. Some evidence suggests that Chinese parents often use technological devices for educational purposes [75], which has the potential to bolster young children’s learning [76]. Still, learning through in-person interactions is critical during the early childhood years [78]. If engaging with screens replaces opportunities for interacting with others face-to-face, such screen exposure may further limit shy young children’s opportunities to develop strong verbal abilities. As such, shy children may struggle to engage in classroom activities.

It should be noted that although shyness was associated with learning difficulties and internalizing problems only at high levels of screen time, shyness was linked to peer problems even when screen exposure was moderate. Although young shy children face a wide range of challenges (including in the academic and emotional domains), these secondary issues are likely rooted in more primary problems related to interpersonal difficulties. For instance, during kindergarten in China, there is a strong focus on socio-emotional learning [79]. As such, young children in China experience pressure to interact socially at school, and failure to engage with teachers and peers may translate to learning problems and poor academic achievement [32,33]. At the same time, shy children may feel lonely and sad because they struggle to interact with peers (despite their desire to engage) during a time when forming friendships is imperative [80,81].

Finally, although screen time was found to moderate links between shyness and certain indicators of socio-emotional functioning (i.e., parent-rated peer problems and teacher-rated internalizing issues), such effects did not emerge for other similar constructs (i.e., parent-rated prosocial behavior and emotional problems). Such discrepancies may be explained (at least in part) by shared method variance. For example, parent-rated shyness was more strongly related to parent-rated emotional problems than to teacher-rated internalizing problems. For prosocial behavior and peer problems, explanations are less clear. Peer problems were assessed using a wide range of constructs, including withdrawn behaviors, friendships, peer-liking, victimization, and dependent relationships with adults. It is thus possible that at least one of these domains differed meaningfully from prosocial behavior. The expansiveness of the peer problems measure may also help explain why even average levels of screen time were linked with peer difficulties. Nevertheless, additional research is required to better understand the present findings.

### 4.2. Implications

Our findings add to the growing research highlighting the negative implications of shyness in young Chinese children [41,42,43]. Indeed, researchers are now developing and evaluating early intervention programs for young shy children in China designed to promote positive peer interactions and reduce social wariness and anxiety [40]. Our findings also provide the first evidence to suggest that increased screen exposure may exacerbate these negative outcomes among shy young children. These results have potentially important implications for shy young children in China. For example, parents of socially withdrawn children are more likely to be overprotective, keeping their children close and limiting free exploration of novel environments [11]. Although parents’ intention here may be to protect their children from feelings of social distress, our findings suggest that it is important to facilitate, encourage, and support peer interactions among shy young children. Recall that screen time exceeded recommended guidelines by almost three times [16]. Rather than allowing shy children to retreat to a virtual world, parents and educators should scaffold and support shy Chinese children’s social interactions to reduce social anxiety and support skill development [40]. Early childhood represents an important and unique developmental period to intervene, as (1) screen time during the early childhood years sets the foundation for lasting habits [22], and (2) parents typically have more control over their children’s technology use. In adolescence, monitoring screen time becomes more difficult [54].

### 4.3. Limitations, Future Directions, and Conclusions

This study addresses a gap in the literature by exploring the links between shyness, screen time, and adjustment among young Chinese children. Notwithstanding, some limitations should be considered in the interpretation of our results, with an eye toward future research directions. First, as noted previously, we did not differentiate between passive and active forms of screen time. However, previous research suggests that at younger ages, screen time more often involves passive activities, such as television and videos [16]. Too much screen time may be especially problematic for shy young children. For example, there is a recent indication that passive technology use may be a particularly negative experience for shy adolescents [82]. Given that researchers have yet to examine how different forms of screen time interact with early childhood shyness to predict adjustment, additional studies may explore whether passive and active forms of screen time have different effects on shyness in young children. In particular, interactive video chatting guided by caring adults may be more beneficial due to social presence [12].

In addition, the present study did not specify screen-related behaviors. For example, specific types of screen behaviors, such as binge-watching (consuming media in rapid succession) and media multitasking (simultaneous use of multiple media platforms, devices, or engagement with a technology-based activity while engaging in a non-technology-based task), appear to be associated with a range of negative consequences, including insomnia, damage to social relationships, academic under-performance, and a sedentary lifestyle [83,84,85,86]. Thus, future research should explore whether similar findings exist for different screen-related behaviors explored in the context of shyness and socio-emotional development among young children.

An additional limitation relates to the screen time measure that was used (i.e., maternal reported screen time over the last week). Previous research suggests that parents misjudge their preschoolers’ technology use retrospectively [87]. To combat recall issues, future research may examine children’s screen exposure using experience sampling or daily diary methodologies. In addition, there is a rising trend in China toward using technology for educational purposes in the classroom [88]. Children may also be exposed to screens when they are in the care of other adults. Thus, parents (and especially only one parent) may not be fully aware of their children’s screen exposure. As such, studies should consider multiple reporter methods, such as teacher, maternal, and paternal reports.

Relatedly, the maternal-rated measure of child behavior problems (SDQ) evidenced poor internal reliability, particularly for the peer problems subscale. Because the SDQ is symptom-based and subscales assess a broad range of behavioral indicators, internal reliabilities for some of the substances do tend to be lower [89]. Future research should include additional parental rating scales of child behavior problems, such as the Child Behavior Checklist (CBC) [90].

Finally, assessing measures at a single time point does not allow us to make conclusions regarding the directions of effects. We must be mindful when interpreting results in the context of previous developmental research, as underlying developmental processes and mechanisms cannot be determined. For example, we have suggested that screen time use exacerbates the associations between shyness and adjustment difficulties. However, it is also possible that shy children who are experiencing greater adjustment difficulties retreat more often to screen time use as an escape from these stressors. Future longitudinal studies are required to better elucidate these effects over time.

Notwithstanding these limitations, results from this study add to our understanding of the role of screen time in the socio-emotional functioning of young shy children. In particular, findings suggest that parents should continue to promote and encourage “face-to-face” peer interactions for young shy children.

## Figures and Tables

**Figure 1 behavsci-13-00763-f001:**
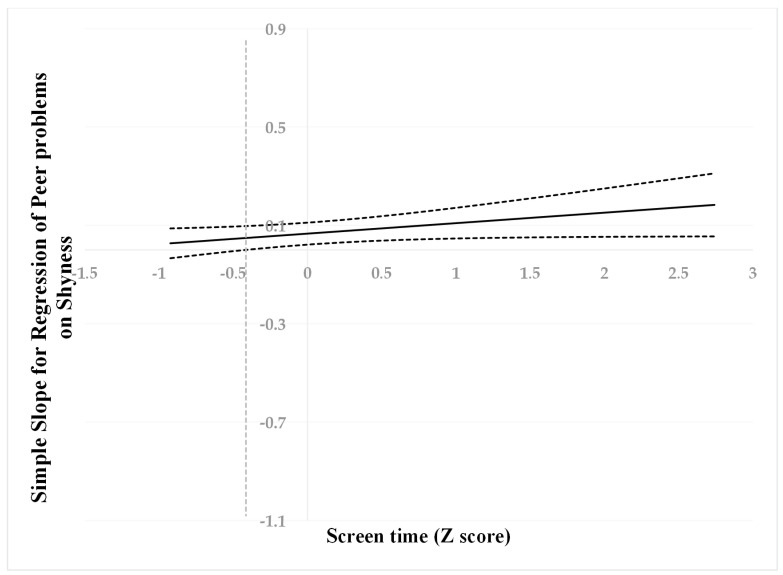
Johnson–Neyman regions of significance and confidence bands for mother-rated shyness along with screen time in relation to peer problems. Solid diagonal line represents the regression coefficient for shyness along screen time. The dashed vertical line is −0.42.

**Figure 2 behavsci-13-00763-f002:**
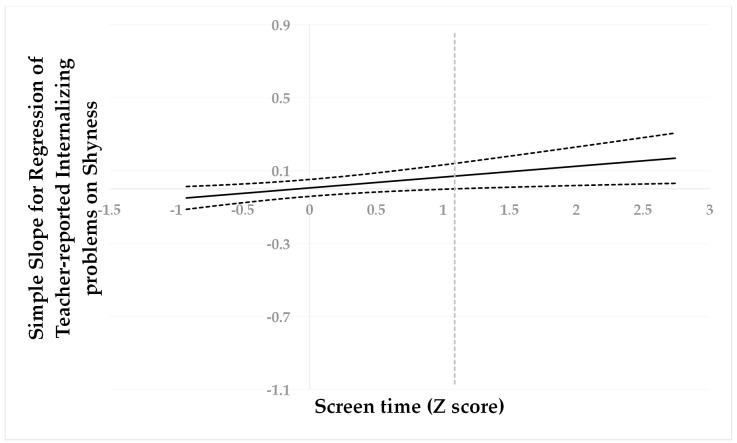
Johnson–Neyman regions of significance and confidence bands for mother-rated shyness along with screen time in relation to internalizing problems. Solid diagonal line represents the regression coefficient for shyness along screen time. The dashed vertical line is 1.09.

**Figure 3 behavsci-13-00763-f003:**
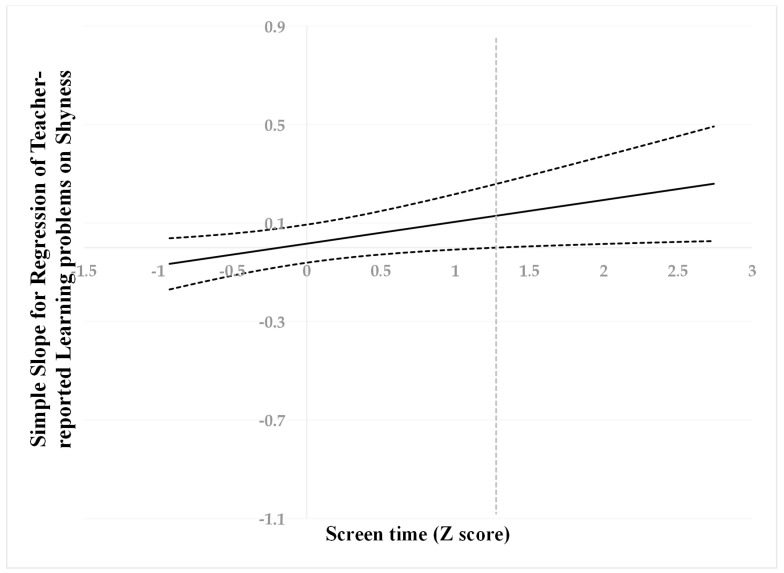
Johnson–Neyman regions of significance and confidence bands for mother-rated shyness along with screen time in relation to learning problems. Solid diagonal line represents the regression coefficient for shyness along screen time. The dashed vertical line is 1.28.

**Table 1 behavsci-13-00763-t001:** Inter-correlations for all study variables (*N* = 208).

	1	2	3	4	5	6	7	8	9
1. parent education^–M^	1								
2. age^–M^	−0.12	1							
3. shyness^–M^	0.14	−0.01	1						
4. screen time^–M^	−0.15 *	0.05	0.05	1					
5. prosocial behavior^–M^	−0.02	0.05	−0.17 *	0.01	1				
6. emotional symptoms^–M^	−0.07	0.05	0.29 ***	0.15 *	−0.13	1			
7. peer problems^–M^	−0.23 **	0.01	0.19 *	0.05	−0.33 ***	0.47 ***	1		
8. internalizing problems^–T^	0.00	0.01	0.08	0.06	0.05	0.06	0.04	1	
9. learning problems^–T^	−0.26 ***	−0.01	0.06	0.12	−0.05	0.11	0.22 **	0.42 ***	1
M	-	58.84	1.85	2.93	2.16	1.43	1.35	0.13	
SD	-	5.32	0.65	3.18	0.51	0.44	0.29	0.33	

* *p* < 0.05, ** *p* < 0.01, *** *p* < 0.001. ^M^ maternal ratings, ^T^ teacher ratings.

**Table 2 behavsci-13-00763-t002:** Effects of shyness, screen time (controlling for parent education and child gender) in relation to indices of social adjustment.

Predictor	B	SE	t-Value	95% CI	*R*²	Δ*R*²	Δ*F*
Prosocial behavior							
Shyness	−0.04	0.03	−1.58	[−0.10, 0.01]	0.04	0.01	0.38
Screen time	0.09	0.03	−0.57	[−0.07, 0.04]			
Shyness × screen time	0.02	0.03	0.62	[−0.03, 0.07]			
Emotional symptoms							
Shyness	0.15	0.03	4.47 ***	[0.08, 0.21]	0.16	0.13	2.38
Screen time	0.06	0.04	1.73	[−0.01, 0.13]			
Shyness × screen time	0.05	0.03	1.52	[−0.02, 0.11]			
Peer problems							
Shyness	0.07	0.03	2.92 **	[0.02, 0.11]	0.11	0.08	3.90
Screen time	−0.01	0.02	−0.37	[−0.06, 0.04]			
Shyness × screen time	0.04	0.02	1.91 ^+^	[0.00, 0.09]			
Internalizing problems							
Shyness	0.01	0.02	0.22	[−0.04, 0.05]	0.07	0.04	6.53
Screen time	0.03	0.03	1.27	[−0.02, 0.08]			
Shyness × screen time	0.06	0.02	2.50 *	[0.01, 0.11]			
Learning problems							
Shyness	0.02	0.04	0.41	[−0.06, 0.09]	0.14	0.11	5.16
Screen time	0.03	0.04	0.62	[−0.06, 0.11]			
Shyness × screen time	0.09	0.04	2.22 *	[0.01, 0.17]			

*CI*, *confidence interval.* ^+^ *p* < 0.10, *
*p* < 0.05, ** *p* < 0.01, *** *p* < 0.001.

## Data Availability

The raw data supporting the conclusions of this article will be made available by the authors without undue reservation.

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
