# Peer review of "Shyness and Socio-Emotional Adjustment among Young Chinese Children: The Moderating Role of Screen Time"

_behavsci, 2023, doi:10.3390/bs13090763_

Round 1

Reviewer 1 Report

Thank you very much for sharing this paper with me. This is an interesting article on the topic of shyness and socio-emotional adjustment among young Chinese children. The authors have succeeded in exploring the moderating role of screen time. I have several minor comments.

The literature review is excellent. The authors define “Shyness” and give clear descriptions of the “Screen Time and Shyness: Diathesis-Stress Model”.

The study would benefit from a more detailed description of the Procedure: When the research was conducted? Why only mothers participated in the research? How the authors got the demographic information for both parents? How the on-line questionnaires were forwarded to the mothers? Was a definition of screen time provided to the mothers?

 The Analytic Strategy section is very helpful. The authors also explain well why they use Johnson-Neyman technique as the particular statistical analysis method.

The study would benefit from discussing limitations about the quite low internal reliability value for the peer problem subscale. What are the authors suggest for future research?

I always prefer to write a General Conclusion after the Limitations section. While the article reads very well, I prefer to end on an upbeat note and would likely add a very short paragraph at the end about the positive contributions this article makes.

Author Response

Reviewer 1

The study would benefit from discussing limitations about the quite low internal reliability value for the peer problem subscale. What are the authors suggest for future research?

Response: Previous studies have demonstrated low internal reliability for the peer problems subscale of the SDQ (Thuenissen et al., 2013), at least in part because the measure is symptom based and considers a diverse range of potential indicators for each subscale. We have added this as a limitation and suggested future research consider other measures of maternal-rated peer problems, such as the Child Behavior Checklist (see p. 14, line 465-466)

I always prefer to write a General Conclusion after the Limitations section. While the article reads very well, I prefer to end on an upbeat note and would likely add a very short paragraph at the end about the positive contributions this article makes.

Response: We have added a short conclusion at the end of the article.

The study would benefit from a more detailed description of the Procedure: When the research was conducted? Why only mothers participated in the research? How the authors got the demographic information for both parents? How the on-line questionnaires were forwarded to the mothers? Was a definition of screen time provided to the mothers?

Response: Thanks for your suggestions. We have added more information in the section of procedure (see the followings; see p.5, line 201-212).

  • The data were collected in February 2019.
  • In China, the phenomenon of the breadwinning father and caregiving mother remains dominant (Li, 2020), and mothers assume the primary caregiving duties (Ji et al., 2017). Therefore, mothers may be able to more accurately report on preschool children’s behavior characteristics. Just as many previous studies also used mother-report on children’s social withdrawal and screen time (Li et al., 2021; Wu et al., 2022; Zhao et al., 2022). Therefore, we used mother-rated data in the present study.
  • The online questionnaire through teachers was forwarded to children’s mothers. In addition, the definition of screen time was presented at the guidance note of the questionnaire (i.e., watching television, using a smartphone, computer, or other digital media device).

And then, we combined peer problems and emotional problems into internalizing problems and revised the manuscript for relevant content.

References

Li, Xin-qi., Yang, Pan-pan., Wang, Wei-jing., & Li, Dan. (2021). The relationship between mother’s punishment and encouragement of sociability and preschool children’s social skill: The moderating effect of shyness. Chinese Journal of Clinical Psychology, 29(1), 98-103. https://doi.org/10.16128/j.cnki.1005-3611.2021.01.020

Li, X. (2020). Fathers’ involvement in Chinese societies: Increasing presence, uneven progress. Child Development Perspectives, 14(3), 150-156. https://doi.org/10.1111/cdep.12375

Ji, Y., Wu, X., Sun, S., & He, G. (2017). Unequal care, unequal work: Toward a more comprehensive understanding of gender inequality in post-reform urban China. Sex Roles, 77, 765-778. https://doi.org/10.1007/s11199-017-0751-1

Wu, Y., Fang, M., Wu, J., Chen, Y., & Li, H. (2022). Shyness and school engagement in Chinese suburban preschoolers: A moderated mediation model of teacher–child closeness and child gender. International Journal of Environmental Research and Public Health, 19(7), 4270. https://doi.org/10.3390/ijerph19074270

Zhao, J., Yu, Z., Sun, X., Wu, S., Zhang, J., Zhang, D., ... & Jiang, F. (2022). Association between screen time trajectory and early childhood development in children in China. JAMA pediatrics, 176(8), 768-775. https://doi.org/10.1001/jamapediatrics.2022.1630

Reviewer 2 Report

This study examined the moderating effect of screen time between shyness and socio-emotional adjustment among young Chinese children and found that for children with higher screen time, the associations between shyness and greater peer problems, internalizing problems and learning problems were higher. The manuscript is clearly written and has important practical implications. Below I raise two suggests to further improve this manuscript.

1. The theoretical and empirical work liking shyness and prosocial behaviors should be reviewed.

2. The reliability for peer problems is too low making it problematic to include peer problems in this study. For typically developing children, the subscales of emotion problems and peer problems are often combined as a measure for internalizing problems. The authors may consider combine emotion problems and peer problems to create a more robust measure. 

Author Response

Reviewer 2

  1. The theoretical and empirical work liking shyness and prosocial behaviors should be reviewed.

 Response: We have added this on p. 2, line 85-90.

  1. The reliability for peer problems is too low making it problematic to include peer problems in this study. For typically developing children, the subscales of emotion problems and peer problems are often combined as a measure for internalizing problems. The authors may consider combine emotion problems and peer problems to create a more robust measure. 

Response: We acknowledge that the internal reliability for the Peer Problems subscale of the SDQ is very low. This is not uncommon for the SDQ and is due, at least in part, to the symptoms-based approach of this measure (please see our reply to the first comment from Reviewer 1). Notwithstanding, we decided not to combine this subscale with the Emotion-Problems subscale. Our rationale for keeping these subscales separate included: (1) peer difficulties and internalizing problems represent distinct domains in terms of outcomes typically studies with relation to shyness; (2) previous studies of shyness among young Chinese children have not utilized this aggregate score (e.g., Baardstu et al., 2022; Hassan et al., 2020; Sette et al., 2016; Sette et al., 2017) and we wanted to allow for direct comparisons in terms of our results.

References

Baardstu, S., Coplan, R. J., Eliassen, E., Brandlistuen, R. E., & Wang, M. V. (2022). Exploring the role of teacher–child relationships in the longitudinal associations between childhood shyness and social functioning at school: A prospective cohort study. School Mental Health, 14(4), 984-996. https://doi.org/10.1007/s12310-022-09518-1

Hassan, R., Poole, K. L., & Schmidt, L. A. (2020). Revisiting the double-edged sword of self-regulation: Linking shyness, attentional shifting, and social behavior in preschoolers. Journal of Experimental Child Psychology, 196, 104842. https://doi.org/10.1016/j.jecp.2020.104842

Sette, S., Baumgartner, E., Laghi, F., & Coplan, R. J. (2016). The role of emotion knowledge in the links between shyness and children's socio‐emotional functioning at preschool. British Journal of Developmental Psychology, 34(4), 471-488. https://doi.org/10.1111/bjdp.12144

Sette, S., Zava, F., Baumgartner, E., Baiocco, R., & Coplan, R. J. (2017). Shyness, unsociability, and socio-emotional functioning at preschool: The protective role of peer acceptance. Journal of Child and Family Studies, 26, 1196-1205. https://doi.org/10.1007/s10826-016-0638-8

Reviewer 3 Report

The authors present a brief but informative study on the association between shyness, socio-emotional adjustment, and screen time. The topic is very hot and meaningful. The study is generally well presented. However, I have noted several issues that require attention before I can recommend this article for publication.

Comment 1:

Social-emotional adjustment might be having a wide definition. I would like to know why the author select the five indicators as the index of social-emotional adjustment. In the introduction part, some explanations to social-emotional adjustment and its index should be added.

Comment 2:

Page 4, line 176, Random sampling might be impossible. If the authors do so, please describe the exact procedure of random sampling.

Comment 3:

The screen time is very critical variable in the present study. But how do authors ensure the accuracy of the measurement to screen time? Only mothers’ report might be not precise. They don’t know children’s screen time in kindergarten. If they are busy and have little time to keep company to their children, they even don’t know the exact screen time in family.

Comment 4:

We only know the Mean and Standard Deviation of the screen time. What’s the implication of them? Is the time of Chinese children exposed to screen high?  

Comment 5:

As the key findings of the present study, the moderated effects should be further discussion. To some indicators of social-emotional adjustment, the moderated effect is not significant. To other indicators, the moderated effect is significant. Why? I appreciate the author's simple effect test with J-N method. If the authors could tell us more about the meaning of the cut-off points of screen time in predicting the simple slopes to different indicators of social-emotional adjustment in J-N method, the article can be more meaningful.

I think the quality of English language is acceptable.

Author Response

Reviewer 3

Comment 1: Social-emotional adjustment might be having a wide definition. I would like to know why the author select the five indicators as the index of social-emotional adjustment. In the introduction part, some explanations to social-emotional adjustment and its index should be added.

Response: As evidenced in our review of the extant literature, there is a robust previous literature linking shyness with maladjustment in three primary domains: (1) internalizing problems; (2) negative peer experiences; and (3) academic difficulties. Accordingly, we selected outcomes in each of these three domains. We have clarified this in the Present Study section (see. p. 4, line171-187).

Comment 2: Page 4, line 176, Random sampling might be impossible. If the authors do so, please describe the exact procedure of random sampling.

Response: We have deleted the “randomly” (pp. 4, line 180).

Comment 3: The screen time is very critical variable in the present study. But how do authors ensure the accuracy of the measurement to screen time? Only mothers’ report might be not precise. They don’t know children’s screen time in kindergarten. If they are busy and have little time to keep company to their children, they even don’t know the exact screen time in family.

Response: First, “screen time” in our study was limited to electronic screen time at home. Secondly, in China, compared to fathers, mothers bear most of the responsibility for child care (Ji et al., 2017). In addition, many previous studies examining preschoolers’ screen time were also reported only by mothers (e.g., Hinkley et al., 2018; Hinkley et al., 2017).

References

Hinkley, T., Brown, H., Carson, V., & Teychenne, M. (2018). Cross sectional associations of screen time and outdoor play with social skills in preschool children. PloS one, 13(4), e0193700. https://doi.org/10.1371/journal.pone.0193700

Hinkley, T., Carson, V., Kalomakaefu, K., & Brown, H. (2017). What mums think matters: a mediating model of maternal perceptions of the impact of screen time on preschoolers' actual screen time. Preventive medicine reports, 6, 339-345. https://doi.org/10.1016/j.pmedr.2017.04.015

Ji, Y., Wu, X., Sun, S., & He, G. (2017). Unequal care, unequal work: Toward a more comprehensive understanding of gender inequality in post-reform urban China. Sex Roles, 77, 765-778. https://doi.org/10.1007/s11199-017-0751-1

Comment 4: We only know the Mean and Standard Deviation of the screen time. What’s the implication of them? Is the time of Chinese children exposed to screen high?  

Response: In our study, the average screen time of 2.93 indicates that children use 2.93 hours per day of screen. According to the Physical activity guideline for Chinese preschoolers recommendations, 1 hour or less per day for children 3 to 6 years of age (Guan et al., 2020). In previous studies, the average screen time of Chinese preschool children was mostly 2.5 hours or less per day (Geng et al., 2023; Hu et al., 2018; Huo et al., 2022; Lan et al., 2020; Li et al., 2022; Xiang et al., 2022; Xie et al., 2020), but some children spent more than 3 hours per day (Hu et al., 2020; Lin et al., 2021). Therefore, our data suggest that the time Chinese children exposed to screens is high in this study.

References

Geng, S., Wang, W., Huang, L., Xie, J., Williams, G. J., Baker, C., ... & Hua, J. (2023). Association between screen time and suspected developmental coordination disorder in preschoolers: A national population-based study in China. Frontiers in Public Health, 11, 1152321. https://doi.org/10.3389/fpubh.2023.1152321

Guan, Hong-yan., Zhao, Xing., Qu, Sha., Wu, Jian-xin., Guo, Jian-jun., & Luo, Dong-mei. (2020). Physical activity guideline for Chinese preschoolers aged 3-6 years. Chinese Journal of Child Health Care, 28(06), 714-720.

Hu, B. Y., Johnson, G. K., & Wu, H. (2018). Screen time relationship of Chinese parents and their children. Children and Youth Services Review, 94, 659-669. https://doi.org/10.1016/j.childyouth.2018.09.008

Hu, B. Y., Johnson, G. K., Teo, T., & Wu, Z. (2020). Relationship between screen time and Chinese children’s cognitive and social development. Journal of Research in Childhood Education, 34(2), 183-207. https://doi.org/10.1080/02568543.2019.1702600

Huo, J., Kuang, X., Xi, Y., Xiang, C., Yong, C., Liang, J., ... & Lin, Q. (2022). Screen time and its association with vegetables, fruits, snacks and sugary sweetened beverages intake among Chinese preschool children in Changsha, hunan province: a cross-sectional study. Nutrients, 14(19), 4086. https://doi.org/10.3390/nu14194086

Lan, Q. Y., Chan, K. C., Kwan, N. Y., Chan, N. Y., Wing, Y. K., Li, A. M., & Au, C. T. (2020). Sleep duration in preschool children and impact of screen time. Sleep Medicine, 76, 48-54. https://doi.org/10.1016/j.sleep.2020.09.024

Li, C., Cheng, G., He, S., Xie, X., Tian, G., Jiang, N., ... & Yan, Y. (2022). Prevalence, correlates, and trajectory of screen viewing among Chinese children in Changsha: a birth cohort study. BMC Public Health, 22(1), 1170. https://doi.org/10.1186/s12889-022-13268-9

Lin, Y. M., Kuo, S. Y., Chang, Y. K., Lin, P. C., Lin, Y. K., Lee, P. H., ... & Chen, S. R. (2021). Effects of parental education on screen time, sleep disturbances, and psychosocial adaptation among Asian preschoolers: a randomized controlled study. Journal of Pediatric Nursing, 56, e27-e34. https://doi.org/10.1016/j.pedn.2020.07.003

Xiang, H., Lin, L., Chen, W., Li, C., Liu, X., Li, J., ... & Guo, V. Y. (2022). Associations of excessive screen time and early screen exposure with health-related quality of life and behavioral problems among children attending preschools. BMC Public Health, 22(1), 1-12. https://doi.org/10.1186/s12889-022-14910-2

Xie, G., Deng, Q., Cao, J., & Chang, Q. (2020). Digital screen time and its effect on preschoolers’ behavior in China: results from a cross-sectional study. Italian Journal of Pediatrics, 46(1), 1-7. https://doi.org/10.1186/s13052-020-0776-x

Comment 5: As the key findings of the present study, the moderated effects should be further discussion. To some indicators of social-emotional adjustment, the moderated effect is not significant. To other indicators, the moderated effect is significant. Why? I appreciate the author's simple effect test with J-N method. If the authors could tell us more about the meaning of the cut-off points of screen time in predicting the simple slopes to different indicators of social-emotional adjustment in J-N method, the article can be more meaningful.

Response: The moderating effects of screen time are discussed in some detail on pages 11-12. We postulated some underlying mechanisms that may help to account for these interaction effects. As well, on page 12 (lines 610-622), we offer some speculation as to why the moderation effects were found for some outcome variables but not others (e.g., shared-method variance). We used Johnson-Neyman (J-N) technique (Johnson & Fay, 1950) to determine the cut-off values. All predictors were standardized for the analyses, then a “region of significance” was estimated for the simple slope of a predictor conditioned on the value of the continuous moderator. The cut-off point is where the “region of non-significance” changes to the “region of significance”.

The results from Figure 1 indicated that when the standardized screen time was higher than -0.92 SD, shyness was positively associated with mother-rated internalizing problems (Figure 1). However, when screen time was lower than -0.92 SD, shyness was no longer associated with mother-rated internalizing problems. When screen time was higher than 1.09 SD, shyness was positively associated with teacher-rated internalizing problems (Figure 2). However, when screen time was lower than 1.09 SD, shyness was no longer associated with internalizing problems. Similarly, when screen time was higher than 1.28 SD, shyness was positively associated with learning problems (Figure 3). However, when screen time was lower than 1.28 SD, shyness was no longer associated with learning problems.

Round 2

Reviewer 2 Report

The authors have adequately addressed my concerns.

The English language is accurate and clear. 

On page 2, in the sentence "It has been suggested that, at least partially, reflect deficits in performance rather than performance, as young shy children’s empthatic and prosocial tendencies might be overwhelmed by feelings of social fear and socio-evaluative concerns", "reflect deficits in performance rather than performance" is confusing. 

Author Response

Thanks for your reviews. We have revised it as "In this regard, young shy children’s empathetic and prosocial tendencies might be overwhelmed by feelings of social fear and socio-evaluative concerns."